# Counterfactual Fairness on Graphs: Augmentations, Hidden Confounders, and Identifiability

## Abstract

We consider augmenting graph data with counterfactual generation in order to achieve fairness on downstream tasks. While this direction has been explored previously, existing methods invariably consider oversimplified causal relationships. Moreover, they often rely on unidentifiable models to encode causal relationships, making it hard to identify the true joint distribution and thus recover counterfactual graphs. To tackle these challenges, we introduce a causal model with hidden confounders on graphs, which considers the existence of hidden confounders affecting both node features and graph structures. We use an identifiable graph VAE model to simultaneously estimate hidden confounders and learn generation functions of the causal model. By incorporating a Gaussian mixture prior distribution, we improve the identifiability of our model to recover the joint distribution of observed data and hidden confounders. Using the generated counterfactual graphs, we enforce consistency in the predictions of classifiers for different counterfactual graphs, thereby achieving graph counterfactual fairness in these classifiers. Experimental results demonstrate the effectiveness of our method in improving the counterfactual fairness of classifiers on various graph tasks. Moreover, theoretical analysis, coupled with empirical results, illustrates the capability of our method to successfully identify hidden confounders.

## 1 Introduction

Graph neural networks (GNNs) have been used to solve problems across a wide range of domains (Gao et al., 2021; Gao & Ji, 2019; Ling et al., 2023a), from social network analysis (Hamilton et al., 2017; Han et al., 2022) to drug discovery (Liu et al., 2022). As with many other machine learning methods, GNNs sometimes make biased predictions (Dai & Wang, 2021; Ling et al., 2023b; Jiang et al., 2022a), thus raising cautions in real-world high-stakes tasks, such as healthcare (Rajkomar et al., 2018), finance (Feldman et al., 2015; Petrasic et al., 2017), and criminal justice (Suresh & Guttag, 2019). Traditional statistical fairness notions (Kang et al., 2020; Dong et al., 2021; Petersen et al., 2021; Jiang et al., 2022b), such as demographic parity (Dwork et al., 2012) and equalized odds (Hardt et al., 2016), have been proposed to quantify and address prediction bias. However, those fairness notions cannot guarantee fair predictions across various dimensions. For example, when demographic parity is equal or close to $0$, it may still allow for unfair treatment of individuals within the same 'protected' category, who may have different characteristics such as varying educational backgrounds or personal attributes. In contrast, counterfactual fairness (Kusner et al., 2017; Wu et al., 2019a;b; Nilforoshan et al., 2022) uses causality to present a more comprehensive and intuitive approach to defining fairness. It allows for the modeling of intricate relationships inherent in real-world data. By considering what would have happened under different conditions, counterfactual fairness offers a different understanding of fairness that can potentially tackle such complex scenarios. Given the potential advantages of counterfactual fairness in addressing the limitations of statistical fairness notions, it is desirable to explore the counterfactual fairness of GNNs to mitigate potential biases.

Indeed, such topics have been explored recently in the literature. Existing methods (Agarwal et al., 2021; Ma et al., 2022) use contrastive learning with counterfactual data augmentations to achieve counterfactual fairness in GNNs. Nevertheless, these methods invariably consider oversimplified

causal relationships. For example, NIFTY (Agarwal et al., 2021) considers a simple causal relationship, where sensitive attributes are unrelated to node features and graph structures. On the other hand, GEAR (Ma et al., 2022) relies on the assumption that the sensitive attribute acts as the sole confounder of node features and graph structures without any hidden confounders. These assumptions may oversimplify the complexities in real-world graph data. Moreover, existing studies use unidentifiable variational autoencoders (VAE) to model causal relationships (Madras et al., 2018; Ma et al., 2022; Louizos et al., 2017; Foster et al., 2022). Due to their non-identifiability (Roeder et al., 2021; Wang et al., 2021; Kivva et al., 2022; Khemakhem et al., 2020a; Foster et al., 2022), it is generally hard to recover the true joint distribution of data. This makes it challenging to accurately recover counterfactual graphs, limiting the effectiveness of these models in achieving counterfactual fairness.

In this work, we introduce graphCFF to address these challenges. We propose a causal model that considers the role of sensitive attributes as confounders that causally affect both node features and graph structures. Moreover, our causal model considers the existence of hidden confounders that impact both node features and graph structures. Consequently, our proposed method provides a more realistic modeling of the underlying complexities in graph data. A primary challenge is that only graph data is observed, while the hidden confounders are unobserved and generation functions of the proposed causal model are unknown. To tackle these challenges, we use an identifiable graph VAE to estimate hidden confounders and learn the generation functions simultaneously. We enhance the identifiability of our model by incorporating a Gaussian mixture model as a prior distribution. This enables our model to accurately recover the joint distribution of observed data and hidden confounders, leading to a more reliable recovery of counterfactuals. We then use the generated counterfactual graphs to enforce consistency in the predictions of classifiers across different counterfactual graphs, thereby achieving graph counterfactual fairness in these classifiers. Experimental results demonstrate the effectiveness of our method in improving the counterfactual fairness of classifiers on various graph tasks. Furthermore, both theoretical analysis and empirical results highlight the potential of our method to successfully recover hidden confounders.

## 2 BACKGROUND AND RELATED WORK

We denote a graph with $n$ nodes as $\mathcal{G} = \{\boldsymbol{A}, \boldsymbol{X}, S\}$. Here, $\boldsymbol{A} \in \{0, 1\}^{n \times n}$ is the adjacency matrix, and $\boldsymbol{A}_{ij} = 1$ if and only if there exists an edge between nodes $i$ and $j$. $\boldsymbol{X} = [X_1, \cdots, X_n]^T \in \mathbb{R}^{n \times p}$ is the node feature matrix, where each $X_i \in \mathbb{R}^p$ is the $p$-dimensional feature vector of node $i$. $S = [s_1, \cdots, s_n]^T \in \{0, 1\}^n$ is the vector containing binary sensitive attributes of nodes, such as gender or race. Here, $s_i$ denotes the sensitive attribute of node $i$. Furthermore, we assume that each node $i$ is associated with a latent variable $Z_i \in \mathbb{R}^q$ and these variables are included in a matrix $\boldsymbol{Z} = [Z_1, \cdots, Z_n]^T \in \mathbb{R}^{n \times q}$. In node classification tasks, the labels of nodes are denoted as $Y = [y_1, \cdots, y_n]^T$, where $y_i \in \{0, 1\}$ is the label of node $i$.

### 2.1 CAUSAL MODELS AND COUNTERFACTUALS

We follow Pearl et al. (2000) to define a causal model as a triple $(\mathcal{U}, \mathcal{V}, \mathcal{F})$, where $\mathcal{V}$ denotes the set of observable variables, $\mathcal{U}$ denotes the set of unobservable variables, and $\mathcal{F} = \{F_1, \cdots, F_{|V|}\}$ denotes the set of functions that assign values to each variable $V_i \in \mathcal{V}$. Any variable $U_i \in \mathcal{U}$ cannot be caused by any variable in $\mathcal{V}$. For each variable $V_i \in \mathcal{V}$, the corresponding function $F_i$ expresses how $V_i$ is dependent on its direct causes as $V_i = F_i(pa_i, U_{pa_i})$, where $pa_i \subseteq \mathcal{V} \setminus \{V_i\}$ represents the observed direct causes of $V_i$, and $U_{pa_i} \subseteq \mathcal{U}$ represents unobserved variables that directly cause $V_i$.

Causal models enable us to perform counterfactual inferences in order to answer questions about what would have happened in a counterfactual world if certain conditions were altered. For example, for two observable variables $V_1$ and $V_2$, a common counterfactual query is "What would the $V_1$ have been if $V_2$ were $V_2'$". The solution to this query, denoted as $V_{1(V_2 \leftarrow V_2')}$, can be obtained through three steps (Glymour et al., 2016) as follows. 1) Abduction: Infer the unobserved variables $\mathcal{U}$ from observed data $\mathcal{V}$. 2) Action: Substitute $V_2$ with $V_2'$. 3) Prediction: Compute the value of remaining element $V_1$ using corresponding function $F_1$ and latent variables $\mathcal{U}$.

## 2.2 COUNTERFACTUAL FAIRNESS ON GRAPHS

Counterfactual fairness is a notion of fairness based on counterfactuals (Kusner et al., 2017; Zuo et al., 2022; Nilforoshan et al., 2022; Wu et al., 2019b; Kilbertus et al., 2017). The prediction of a classifier for an individual is counterfactually fair if the prediction remains consistent in both the actual and the counterfactual worlds, where the individual belongs to a different demographic group. Recent studies have explored counterfactual fairness on graph data (Agarwal et al., 2021; Ma et al., 2022). These studies aim to develop a counterfactually fair encoder to ensure that the learned representations from the original graph and those of the counterfactual graphs are identical. NIFTY proposes to generate counterfactual graphs by flipping the sensitive attributes of nodes while leaving other node features and graph structures unchanged (Agarwal et al., 2021). A triplet-based objective is then used to maximize agreement between the original and counterfactual graphs in order to learn counterfactually fair representations. However, a major limitation of NIFTY is its focus on a simplified case, where the sensitive attribute has no causal effect on other node features or graph structures. Instead of ignoring all causal relationships, GEAR addresses the case where sensitive attributes are the cause of both node features and graph structures (Ma et al., 2022). GEAR uses VAEs to model these causal relationships and generate counterfactual graphs by perturbing sensitive attributes. A Siamese network is then used to minimize discrepancies between representations learned from the original graph and counterfactual graphs, leading to counterfactually fair representations. Nevertheless, GEAR operates under the assumption that there are no hidden confounders (Pearl, 2009; Greenland et al., 1999; Mickey & Greenland, 1989) of node features and graph structures, which could oversimplify the complexities of real-world graph data. Furthermore, VAE models are generally non-identifiable (Khemakhem et al., 2020a; Kivva et al., 2022; Wang et al., 2021; Foster et al., 2022), posing challenges in accurately recovering the true latent variables of the causal model and thus generating noisy counterfactual data.

## 3 METHODOLOGY

While previous studies focus on the counterfactual fairness of encoders (Ma et al., 2022; Agarwal et al., 2021), we consider the counterfactual fairness of classifiers in this work. We propose GraphCFF, a novel method to achieve counterfactual fairness by augmenting graph data with counterfactual generation. In this section, we first conceptualize and describe our proposed causal model, which takes into account the existence of hidden confounders. We then introduce how we use variational inference to learn both a generative model, which approximates the functions of the causal model, and an inference model, which estimates hidden confounders. We next explain how we generate counterfactual graphs by flipping the sensitive attributes of nodes. Using these generated counterfactual graphs, we are able to train a classifier that achieves counterfactual fairness.

We provide a formal definition of graph counterfactual fairness as follows:

**Definition 1** (Graph Counterfactual Fairness). Given a graph $\mathcal{G} = \{\boldsymbol{A}, \boldsymbol{X}, S\}$, the prediction $\hat{Y}$ of a classifier is counterfactually fair if, for any node $i$,

$$P(\hat{Y}_i | \boldsymbol{A}, \boldsymbol{X}, S) = P(\hat{Y}_{i(S \leftarrow S')} | \boldsymbol{A}, \boldsymbol{X}, S) \tag{1}$$

where

$$s'_j = \begin{cases} s_j, & i \neq j \\ 1 - s_j, & i = j \end{cases} \quad \text{for } j = 1, \cdots, n,$$

and $\hat{Y}_i$ is the prediction for node $i$.

In other words, the prediction of a classifier for a node is counterfactually fair if it remains the same in both the actual world and the counterfactual world where the sensitive attribute of this node is assigned to a different demographic group. Note that obtaining counterfactual data in the real world is almost impossible. Moreover, the causal relationships between the sensitive attribute $S$, node features $\boldsymbol{X}$, and adjacency matrix $\boldsymbol{A}$ remain unknown, making it hard to perform counterfactual inference directly. Therefore, how to obtain counterfactuals is one of the primary challenges to be addressed in counterfactual fairness on graphs.

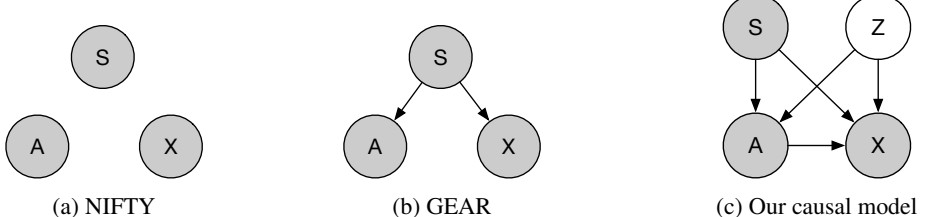

(a) NIFTY        (b) GEAR        (c) Our causal model

Figure 1: Comparison of different approaches for modeling the causal relations among sensitive attributes $S$, node features $\boldsymbol{X}$, and adjacency matrix $\boldsymbol{A}$. Grey and white nodes represent observed and unobserved variables, respectively. (a) NIFTY (Agarwal et al., 2021) assumes that sensitive attribute has no causal effect on other node features or adjacency matrix. (b) GEAR (Ma et al., 2022) models sensitive attributes $S$ as the sole confounder of node features $\boldsymbol{X}$ and adjacency matrix $\boldsymbol{A}$ without any hidden confounders. (c) GraphCFF (ours) explicitly models hidden confounders $\boldsymbol{Z}$ for both node features $\boldsymbol{X}$ and adjacency matrix $\boldsymbol{A}$.

### 3.1 CAUSAL MODEL WITH HIDDEN CONFOUNDERS

In this section, we present our proposed causal model, accounting for the presence of hidden confounders in graph data such as social networks. We first introduce independent latent factors $\boldsymbol{Z}$, which generate the observed graph data. The latent factors of different nodes are independent and identically distributed. We assume that the latent factors $\boldsymbol{Z}$ serve as hidden confounders for both node features $\boldsymbol{X}$ and adjacency matrix $\boldsymbol{A}$. For example, an individual's 'personality' could influence his/her 'health' and his/her social relationships with others. Since $\boldsymbol{Z}$ does not necessarily correspond to tangible objects in the world, it is reasonable to consider hidden confounders as independent of sensitive attributes in our model. Moreover, we assume sensitive attributes $S$ have no parent variables in the causal graph, implying that they can only be the cause of other variables. Specifically, sensitive attributes $S$ are not caused by $\boldsymbol{Z}$, while they are the cause of both node features $\boldsymbol{X}$ and adjacency matrix $\boldsymbol{A}$. In fairness problems, sensitive attributes are typically determined at birth, making it reasonable to treat them as root nodes in a causal graph. For example, the sensitive attribute 'gender' cannot be caused by other features like 'height', while 'gender' can cause 'height'. This assumption has been widely adopted in recent studies (Kusner et al., 2017; Zuo et al., 2022; Ma et al., 2022; Madras et al., 2019).

To model the topological structures of graphs, we assume that the edge between node $i$ and node $j$ only depends on the factors of these two nodes and is independent of other nodes. We also explicitly account for sensitive homophily in graphs, indicating that nodes with the same sensitive attribute are more likely to connect than nodes with different sensitive attributes. In other words, our causal model is similar to stochastic block models with higher intra-connection and lower inter-connection probabilities. This is consistent with recent studies (Jiang et al., 2022a; Dong et al., 2022; Kose & Shen, 2022; Dai & Wang, 2021). Formally, the process of generating the adjacency matrix $\boldsymbol{A} = F_A(\boldsymbol{Z}, S)$ can be expressed as

$$\boldsymbol{A}_{ij} = \begin{cases} 1, & \text{if } s_i \neq s_j \text{ and } \text{sim}(Z_i, Z_j) > 1 - P_{\text{inter}} \\ 1, & \text{if } s_i = s_j \text{ and } \text{sim}(Z_i, Z_j) > 1 - P_{\text{intra}} \\ 0, & \text{otherwise} \end{cases} \quad \text{for } i, j = 1, \cdots, n, \quad (2)$$

where $\text{sim}(\cdot)$ computes the similarity between two nodes. $P_{\text{inter}}$ and $P_{\text{intra}}$ represent thresholds for the intra-connection and inter-connection, respectively, with $P_{\text{inter}} < P_{\text{intra}}$.

Furthermore, one of the most significant challenges in graph data is the non-independence of nodes, which means altering one node could affect another node. For example, if all friends of an individual adopt a diet, it is likely that the individual will also start a diet. Instead of treating each node as an independent data point (Agarwal et al., 2021), we explicitly consider the causal relationship between nodes, meaning that the node features of node $i$ can be influenced by its neighbors. We rely on a widely accepted local-dependence assumption for graph-structured data (Wu et al., 2020); that is, given data related to neighbors within a certain number of hops of node $i$, the data in the rest of the graph is independent of node $i$. In other words, a node depends on its $k$-hop neighbors but is independent of nodes outside its $k$-hop neighborhood. In this work, we consider cases where $k = 1$, which should provide a good approximation of real-world scenarios. Formally, the generation

process of node features $\boldsymbol{X} = F_X(\boldsymbol{Z}, \boldsymbol{A}, S)$ can be expressed as

$$X_i = F_{X_1}(Z_i, s_i) + \text{AGG}(\{F_{X_2}(Z_j, s_j) : j \in N_i\}) \quad \text{for } i = 1, \cdots, n, \tag{3}$$

where $N_i$ denotes the set of the nodes connected to the node $i$ in the graph, and $\text{AGG}(\cdot)$ denotes an aggregation function that maps the messages from all neighboring nodes to a single vector. $F_{X_1}$ and $F_{X_2}$ are two piecewise affine transformation functions, such as multilayer perceptrons with leaky ReLU activations. In summary, our proposed causal model can be formally defined as follows:

**Definition 2** (Causal Model with Hidden Confounders on Graphs)**.** A causal model on graphs with hidden confounders can be represented as a tuple $(\mathcal{U}, \mathcal{V}, \mathcal{F})$, where $\mathcal{V} \equiv \{\boldsymbol{A}, \boldsymbol{X}, S\}$ and $\mathcal{U} \equiv \{\boldsymbol{Z}\}$. $\mathcal{F} = \{F_A, F_X\}$ corresponds to the generating mechanism of $\boldsymbol{A}$ and $\boldsymbol{X}$, defined as $\boldsymbol{A} = F_A(\boldsymbol{Z}, S)$ and $\boldsymbol{X} = F_X(\boldsymbol{Z}, \boldsymbol{A}, S)$.

*Remark* 3.1. Different from the simple scenario (Ma et al., 2022), where $S$ is the only confounder of $\boldsymbol{A}$ and $\boldsymbol{X}$, and no hidden confounders exist, we consider a more realistic scenario where hidden confounders of $\boldsymbol{A}$ and $\boldsymbol{X}$ exist, which has been overlooked in the literature. See Figure 1 for a comparison between our proposed causal model with the causal models in existing works. More discussions are given in Appendix C.

## 3.2 LEARNING METHOD

A primary challenge in performing counterfactual inferences on our causal model lies in the fact that only graph data $\mathcal{G} = \{\boldsymbol{A}, \boldsymbol{X}, S\}$ is observed, while the hidden confounders $\boldsymbol{Z}$ are unobserved and generation functions $\mathcal{F}$ are unknown. As demonstrated by Louizos et al. (2017); Madras et al. (2019), counterfactuals can be recovered if we can recover the joint distribution $P(\boldsymbol{Z}, \boldsymbol{A}, \boldsymbol{X}, S)$ among node features $\boldsymbol{X}$, graph structures $\boldsymbol{A}$, sensitive attributes $S$, and the hidden confounders $\boldsymbol{Z}$. Therefore, we need to learn two models; namely (1) a generative model approximating the generation functions $\mathcal{F}$, and (2) an inference model approximating the distribution of $\boldsymbol{Z}$ given $\boldsymbol{A}, \boldsymbol{X}, S$.

We use variational inference parameterized by deep neural networks to learn the parameters of both models jointly. In variational inference, we aim to learn an approximation of the joint distribution $P(\boldsymbol{Z}, \boldsymbol{A}, \boldsymbol{X}, S)$, by maximizing a variational lower bound on the log-probability of the observed graph data $(\boldsymbol{A}, \boldsymbol{X}, S)$. Specifically, we introduce a variational distribution $Q(\boldsymbol{Z}|\boldsymbol{A}, \boldsymbol{X}, S)$ which uses a parametric family of distributions to approximate the intractable posterior distribution $P(\boldsymbol{Z}|\boldsymbol{A}, \boldsymbol{X}, S)$. The parameters of both the generative model and the inference model are learned simultaneously by maximizing the evidence lower bound (ELBO) of the marginal data likelihood.

To learn a generative model that is consistent with our proposed causal model, we model the joint probability $P(\boldsymbol{Z}, \boldsymbol{A}, \boldsymbol{X}, S)$ using the following factorization:

$$P(\boldsymbol{Z}, \boldsymbol{A}, \boldsymbol{X}, S) = P(\boldsymbol{Z})P(S)P(\boldsymbol{A}|\boldsymbol{Z}, S)P(\boldsymbol{X}|\boldsymbol{A}, \boldsymbol{Z}, S), \tag{4}$$

where $P(\boldsymbol{A}|\boldsymbol{Z}, S)$ and $P(\boldsymbol{X}|\boldsymbol{A}, \boldsymbol{Z}, S)$ correspond to the generation functions $F_A$ and $F_X$, respectively. To learn the parameters of both the generative model and the inference model, we maximize the ELBO as

$$\begin{aligned}
\log P(\boldsymbol{A}, \boldsymbol{X}|S) \geq & \mathbb{E}_{Q(\boldsymbol{Z}|\boldsymbol{A}, \boldsymbol{X}, S)}[\log P(\boldsymbol{A}|\boldsymbol{Z}, S) + \log P(\boldsymbol{X}|\boldsymbol{A}, \boldsymbol{Z}, S) \\
& + \log P(\boldsymbol{Z}) - \log Q(\boldsymbol{Z}|\boldsymbol{A}, \boldsymbol{X}, S)].
\end{aligned} \tag{5}$$

We provide detailed derivations of ELBO in Appendix B. The ELBO can be maximized using the Stochastic Gradient Variational Bayes estimator and the reparameterization trick (Kingma & Welling, 2014).

## 3.3 GAUSSIAN MIXTURE AS PRIOR DISTRIBUTION

It is important to note that the conclusion by Louizos et al. (2017); Madras et al. (2019) is based on the assumption that learning with VAEs can recover the true joint distribution $P(\boldsymbol{Z}, \boldsymbol{A}, \boldsymbol{X}, S)$. However, due to the non-identifiability of VAE models (Khemakhem et al., 2020a; Wang et al., 2021; Foster et al., 2022), it is generally impossible to recover the true joint distribution $P(\boldsymbol{Z}, \boldsymbol{A}, \boldsymbol{X}, S)$. This non-identifiable issue undermines the conclusion by Louizos et al. (2017); Madras et al. (2019). To address this issue, we follow Kivva et al. (2022) to incorporate a Gaussian mixture model (GMM) prior in our framework. We use the same GMM to model the prior distribution of hidden confounders $Z_i$ for any node $i$. Specifically, we model the prior distribution of the hidden

confounders $Z_i$ as a weighted sum of $K$ Gaussian components, where each component $c$ has a mixture weight $\pi_c \in \mathbb{R}^+$. These mixture weights are themselves probabilities that satisfy the constraints $\sum_{c=1}^{K} \pi_c = 1$ and $\pi_c > 0$. Thus, the probabilities of the components can be modeled using a categorical distribution as $P(c) \sim \text{Cat}(\pi_1, \ldots, \pi_K)$. Each Gaussian component is parameterized by a mean vector $\mu_c \in \mathbb{R}^q$ and a covariance matrix $\boldsymbol{\Sigma_c} \in \mathbb{R}^{q \times q}$. The covariance matrix $\boldsymbol{\Sigma_c}$ is a diagonal matrix with diagonal elements $\sigma_{c,1}^2, \sigma_{c,2}^2, \ldots, \sigma_{c,q}^2$. Formally, the prior distribution of the hidden confounders $Z_i$ can be described as $Z_i \sim \sum_{c=1}^{K} \pi_c \mathcal{N}(\mu_c, \boldsymbol{\Sigma_c})$, where $\mathcal{N}(\mu_c, \boldsymbol{\Sigma_c})$ denotes a Gaussian distribution with mean $\mu_c$ and covariance matrix $\boldsymbol{\Sigma_c}$. Let $C$ denote the vector containing the component values of all nodes in the graph. The updated ELBO of our framework can be formally described as

$$
\begin{aligned}
\log P(\boldsymbol{A}, \boldsymbol{X}|S) \geq & \mathbb{E}_{Q(\boldsymbol{Z}, C|\boldsymbol{A}, \boldsymbol{X}, S)}[\log P(\boldsymbol{A}|\boldsymbol{Z}, S) + \log P(\boldsymbol{X}|\boldsymbol{A}, \boldsymbol{Z}, S) + \log P(\boldsymbol{Z}|C) \\
& + \log P(C) - \log Q(\boldsymbol{Z}|\boldsymbol{A}, \boldsymbol{X}, S) - \log Q(C|\boldsymbol{A}, \boldsymbol{X}, S)].
\end{aligned}
\tag{6}
$$

Note that we assume the variational distribution $Q(\boldsymbol{Z}, C|\boldsymbol{A}, \boldsymbol{X}, S)$ can be factorized as $Q(\boldsymbol{Z}, C|\boldsymbol{A}, \boldsymbol{X}, S) = Q(\boldsymbol{Z}|\boldsymbol{A}, \boldsymbol{X}, S)Q(C|\boldsymbol{A}, \boldsymbol{X}, S)$. This assumption is also used in VaDE (Jiang et al., 2016). By incorporating the GMM prior, we enhance the ability of our model to recover the distribution of the hidden confounders, resulting in accurate counterfactual inferences on our model. Besides, we provide theoretical support for identifying the true distribution of the hidden confounders in Section 4.

### 3.4 Counterfactual Generation

In this section, we discuss the process of generating counterfactual graphs using well-trained generative and inference models, enabling the learning of counterfactually fair downstream classifiers. Once the learning process is complete, the generative model can generate new graph data consistent with the generation mechanism as our proposed causal model. Meanwhile, the inference model can estimate the hidden confounders given the observed graph data. The process of generating a counterfactual graph, in which the sensitive attribute of node $i$ is assigned to a different group, consists of three main steps. Given the observed graph data $\mathcal{G} = \{\boldsymbol{A}, \boldsymbol{X}, S\}$, we first estimate the hidden confounders $\boldsymbol{Z}$ using the inference model. Next, we swap the sensitive attribute of node $i$ to create a new matrix $S'$, where

$$
s_j' = \begin{cases} s_j, & i \neq j \\ 1 - s_j, & i = j \end{cases} \quad \text{for } j = 1, \cdots, n.
$$

With the obtained hidden confounders $\boldsymbol{Z}$ and the new sensitive attribute matrix $S'$, we use the generative model to generate the counterfactual graph $\mathcal{G}_{(S \leftarrow S')} = \{\boldsymbol{A}_{(S \leftarrow S')}, \boldsymbol{X}_{(S \leftarrow S')}, S'\}$, where $\boldsymbol{A}_{(S \leftarrow S')} = F_A(\boldsymbol{Z}, S')$ and $\boldsymbol{X}_{(S \leftarrow S')} = F_X(\boldsymbol{Z}, \boldsymbol{A}_{(S \leftarrow S')}, S')$.

The generated counterfactual graphs can then be used to train counterfactually fair downstream classifiers. Intuitively, we augment the observed graph dataset with counterfactual graphs while keeping the labels the same, ensuring that downstream classifiers make consistent predictions for different counterfactual graphs. This encourages the graph counterfactual fairness in the downstream classifiers. In practice, during each training step, we randomly select a node $i$ and generate a counterfactual graph where the sensitive attribute of node $i$ is flipped. Along with the node labels $Y$, the generated counterfactual graph $\mathcal{G}_{(S \leftarrow S')}$ is used to train the downstream classifiers.

### 4 Identifiability Analysis

As we mentioned in Section 3.3, one primary issue of learning with VAE models is that they are non-identifiable. In other words, multiple parameter sets can result in models having identical marginal data and prior distributions, but entirely different latent factors $\boldsymbol{Z}$. This makes it hard to recover the true joint distribution $P(\boldsymbol{Z}, \boldsymbol{A}, \boldsymbol{X}, S)$ and, in turn, makes it difficult to recover counterfactuals. Therefore, ensuring the identifiability of our model becomes important for recovering counterfactuals. A model is considered identifiable if there is only one unique set of parameter values corresponding to the true data-generating process. Nevertheless, this strict definition of identifiability may be overly restrictive in practical applications. Instead, we consider a relaxed notion of identifiability. Following Kivva et al. (2022), we provide a formal definition of identifiability as follows:

**Definition 3** (Identifiability). Let $\mathfrak{P}$ be a family of probability distributions on the latent factors and $\mathfrak{F}$ be a family of the generation functions. We use $F_\sharp P$ to denote the pushforward measure of $P$ by $F$. For a pair $(P, F) \in \mathfrak{P} \times \mathfrak{F}$, we say that the distribution $P$ is identifiable from $F_\sharp P$ up to an affine transformation if for any $(P', F') \in \mathfrak{P} \times \mathfrak{F}$ such that $F_\sharp P \equiv F'_\sharp P'$ there exists an invertible affine transformation $h$ such that $P' = h_\sharp P$.

This identifiability definition can be easily extended to other invertible transformations, such as permutations and translations. See Khemakhem et al. (2020a;b); Roeder et al. (2021) for different notions of identifiability in deep latent variable models. To show the identifiability of our model, we make a mild assumption on the generating function $F$, defined as follows.

**Definition 4** (Weakly Injective Function). Let $B(x, \delta)$ denote a ball centered at a point $x$ and with a radius of $\delta$. We say that a function $F : \mathbb{S} \to \mathbb{T}$ is weakly injective if (i) there exists $x_0 \in \mathbb{T}$ and $\delta > 0$ s.t. $|F^{-1}(\{x\})| = 1$ for every $x \in B(x_0, \delta) \cap F(\mathbb{S})$, and (ii) $\{x : |F^{-1}(\{x\})| = \infty\} \subseteq F(\mathbb{S})$ has measure zero with respect to the Lebesgue measure on $F(\mathbb{S})$.

*Remark* 4.1. The assumption that functions are weakly injective is indeed a weaker condition than the assumption that functions are injective. Note that ReLU network networks are generically weakly injective under simple assumptions on their architecture. See Appendix H of Kivva et al. (2022) for more details.

With a mild assumption that the generation function $F$ is weakly injective, we show the identifiability of our model as the following Theorem.

**Theorem 1.** Let $F$ be the generation function of our proposed causal model with hidden confounders on graphs, defined as $F(\boldsymbol{Z}, S) = (\boldsymbol{A}, \boldsymbol{X}) = (F_A(\boldsymbol{Z}, S), F_X(\boldsymbol{Z}, F_A(\boldsymbol{Z}, S), S))$. If $F$ is weakly injective, then we have $P(\boldsymbol{Z}, C)$ to be identifiable from $P(\boldsymbol{A}, \boldsymbol{X}, S)$ up to permutation, scaling, and/or translation.

The complete proof of this theorem is given in Appendix A. Theorem 1 shows that the learned hidden confounders can recover the true ones up to a certain degree. This implies that we can better recover the ground truth of the joint distribution $P(\boldsymbol{Z}, \boldsymbol{A}, \boldsymbol{X}, S)$ among node features $\boldsymbol{X}$, graph structures $\boldsymbol{A}$, sensitive attributes $S$, and the hidden confounders $\boldsymbol{Z}$, enabling us to recover the true counterfactual graphs. See Section 5.3 for empirical studies.

## 5  EXPERIMENTS

In this section, we evaluate the proposed method on both synthetic and real-world graphs. Experimental results show that our method outperforms many baselines on node classification tasks in terms of both fairness and accuracy. Furthermore, we conduct experiments on synthetic datasets to verify the identifiability of our model. More experimental results are given in Appendix E.

### 5.1  EXPERIMENTAL SETTINGS

**Setup.** To compare our method with other baselines, we train a GCN (Kipf & Welling, 2017) based classification model with different augmentations. We evaluate the performance of our method and other baselines by the averaged testing accuracy and counterfactual fairness of the classification model over five runs. For a fair comparison, we use the same architecture of the classification model. See more experimental details in Appendix D.

**Evaluation Metrics.** We use accuracy to evaluate the prediction performance of the classification model on node classification tasks. To quantify counterfactual fairness, we define the metric as

$$\Delta_{\mathrm{CF}} = \frac{1}{n} \sum_i |P(\hat{Y}_i | \boldsymbol{A}, \boldsymbol{X}, S) - P(\hat{Y}_{i(S \leftarrow S^{(i)})} | \boldsymbol{A}, \boldsymbol{X}, S)| \tag{7}$$

where $S^{(i)}$ denotes the matrix obtained by flipping the sensitive attribute of node $i$, and $\hat{Y}_i$ is the prediction for node $i$. See more evaluation metrics in Appendix E.1.

**Baselines.** We compare our method with the following baseline methods, including (1) No augmentation, which only uses the original graph to train the downstream classifier; (2) NIFTY (Agarwal et al., 2021), which creates counterfactual graphs by flipping the sensitive attribute while leaving

Table 1: Comparisons between our method and baselines on node classification tasks in terms of accuracy and counterfactual fairness. The best results are shown in bold.

| Dataset | No augmentation | | NIFTY | | GEAR | | GraphCFF | |
|---|---|---|---|---|---|---|---|---|
| | ACC ↑ | $\Delta_{CF}$ ↓ | ACC ↑ | $\Delta_{CF}$ ↓ | ACC ↑ | $\Delta_{CF}$ ↓ | ACC ↑ | $\Delta_{CF}$ ↓ |
| Synthetic Linear | $90.00 \pm \mathbf{1.41}$ | $9.63 \pm 0.61$ | $88.20 \pm 0.47$ | $6.63 \pm 0.46$ | $85.60 \pm 2.05$ | $4.60 \pm 1.56$ | $88.20 \pm 0.47$ | $\mathbf{3.47 \pm 0.19}$ |
| Synthetic NonLinear | $89.00 \pm 0.82$ | $14.27 \pm 0.98$ | $89.50 \pm 0.47$ | $6.57 \pm 0.29$ | $86.20 \pm 1.41$ | $6.70 \pm 1.81$ | $\mathbf{90.00 \pm 0.82}$ | $\mathbf{2.60 \pm 0.43}$ |
| Synthetic Noise | $\mathbf{92.60 \pm 0.94}$ | $7.17 \pm 1.35$ | $91.6 \pm 0.47$ | $6.40 \pm 0.64$ | $85.40 \pm 4.03$ | $4.33 \pm 1.71$ | $90.40 \pm 1.25$ | $\mathbf{2.47 \pm 0.90}$ |
| German | $68.00 \pm 1.41$ | $18.93 \pm 1.75$ | $67.00 \pm 0.82$ | $8.67 \pm 1.76$ | $67.60 \pm 1.89$ | $6.60 \pm 2.44$ | $\mathbf{68.30 \pm 1.70}$ | $\mathbf{1.50 \pm 0.70}$ |
| Credit | $\mathbf{79.80 \pm 0.34}$ | $28.80 \pm 3.45$ | $78.02 \pm 2.46$ | $7.83 \pm 3.86$ | $79.50 \pm 1.94$ | $12.94 \pm 5.64$ | $77.20 \pm 1.46$ | $\mathbf{1.76 \pm 0.32}$ |
| Bail | $\mathbf{94.40 \pm 0.2}$ | $65.49 \pm 0.12$ | $94.16 \pm 0.35$ | $65.32 \pm 0.26$ | $88.77 \pm 2.45$ | $50.15 \pm 10.27$ | $87.76 \pm 3.5$ | $\mathbf{28.95 \pm 5.27}$ |

other node features and graph structures unchanged; (3) GEAR (Ma et al., 2022), which uses a graph auto-encoder trained with fairness constraints to generate counterfactuals. Note that both NIFTY and GEAR use contrastive objectives to ensure counterfactual fairness of the learned representations, but we only use the counterfactual generation part of both approaches to augment datasets and train the downstream classifier.

**Synthetic Datasets.** We use our proposed causal model to create three synthetic datasets, namely Synthetic Linear, Synthetic NonLinear, and Synthetic Noise. The value of hidden confounders $\boldsymbol{Z}$, sensitive attribute $S$, the adjacency matrix $\boldsymbol{A}$ and node feature $\boldsymbol{X}$ are computed by ancestral sampling. In Synthetic Linear, $F_{X_1}$ and $F_{X_2}$ are two full-rank linear transformations. For Synthetic NonLinear, we use random initialized multi-layer perceptrons (MLPs) for $F_{X_1}$ and $F_{X_2}$. Synthetic Noise is generated by randomly flipping adjacency matrix entries and adding exponential noise to node features. Each dataset is randomly partitioned into training, validation, and test sets, at proportions of $80\%$, $10\%$, and $10\%$, respectively. In the synthetic datasets, we can fully manipulate the data generation process and thus easily generate the counterfactual graphs. For each node, we flip its sensitive attribute to get a new sensitive attribute vector $S'$. Counterfactual graphs are then generated as $\mathcal{G}_{(S \leftarrow S')} = \{\boldsymbol{A}_{(S \leftarrow S')}, \boldsymbol{X}_{(S \leftarrow S')}, S'\}$, where $\boldsymbol{A}_{(S \leftarrow S')} = F_A(\boldsymbol{Z}, S')$ and $\boldsymbol{X}_{(S \leftarrow S')} = F_X(\boldsymbol{Z}, \boldsymbol{A}_{(S \leftarrow S')}, S')$. See Appendix D.1 for more details.

**Real-world Datasets.** We further demonstrate the advance of our method using three real-world datasets (Agarwal et al., 2021), including German, Credit, and Bail. The German dataset uses gender as the sensitive attribute, and the task is to predict the credit risk status of clients, categorizing them as either good or bad risks. Next, the Credit dataset uses age as the sensitive attribute and the task is to predict whether an individual is likely to default on a credit card payment. Finally, the Bail dataset uses race as the sensitive attribute, and the task is to predict whether defendants will bail, i.e., the individuals who are less likely to commit a violent crime if released. For all three datasets, we randomly split $80\%/10\%/10\%$ for training, validation, and test datasets. Since the ground-truth counterfactual is unknown, we generate the counterfactual data based on the causal discovery methods implemented in Tetrad (Ramsey et al., 2018). See more details in Appendix D.1.

## 5.2 EXPERIMENTAL RESULTS

**Fairness and Accuracy Performance.** Table 1 shows accuracy and counterfactual fairness metrics of our method, compared with baselines in Section 5.1 on both the synthetic datasets and the real-world datasets. Our method consistently achieves the best counterfactual fairness performance on all six datasets. For example, compared to only using the observed graph data without any augmentations, our method reduces counterfactual fairness by $63.9\%$, $81.8\%$, and $65.6\%$ on Synthetic Linear, Synthetic NonLinear, and Synthetic Noise, respectively, with comparable accuracy performance. It is worth noting that in some cases, our method can improve counterfactual fairness with little sacrifice to accuracy performance, or even enhance the accuracy performance. For example, compared to no augmentations, our method can achieve a higher accuracy performance on Synthetic NonLienar and German datasets. Such observations demonstrate the advance of our proposed method.

**Fairness and Accuracy Trade-off.** We further compare the fairness and accuracy trade-off performance of our method with baselines on both the synthetic datasets and the real-world datasets. As shown in Figure 2, we create Pareto front curves by assigning different weights to the counterfactual graphs generated by each method during the training process of the classifier model. The upper-left corner point represents the ideal performance, i.e., the highest accuracy and lowest prediction bias. The results show that our method achieves the best ACC-CF trade-off compared with baselines on six datasets.

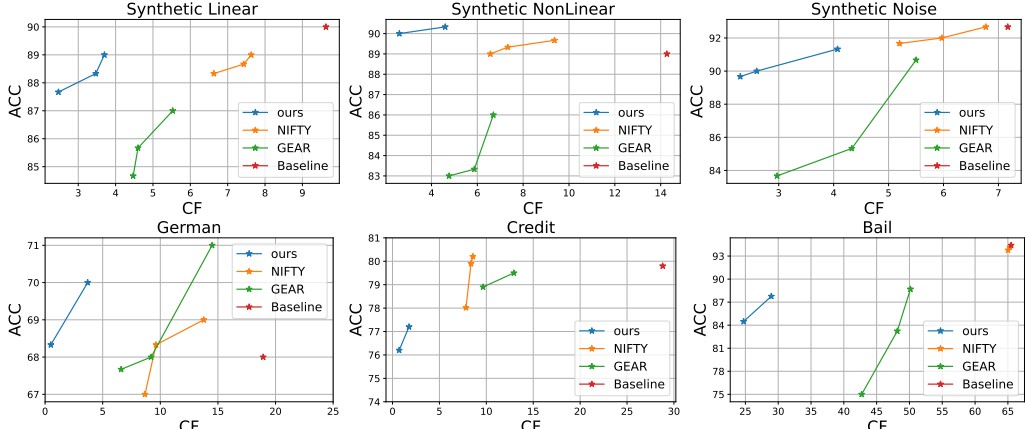

Figure 2: Accuracy and counterfactual fairness trade-off on the six datasets. A better trade-off is achieved when the performance appears in the upper left corner where higher accuracy and lower prediction bias are obtained simultaneously.

## 5.3 Identifiability Evaluation

In this section, we investigate the identifiability of our model. We conduct experiments to support our theoretical analysis in Section 4. To quantify identifiability, we use the mean correlation coefficient (MCC) metric, as in previous studies (Khemakhem et al., 2020b; Kivva et al., 2022; Khemakhem et al., 2020a). Given two multivariate random variables, the MCC metric calculates the

Table 2: Identifiability study of our model.

| Dataset | GraphCFF w/o GMM | | GraphCFF | |
|---|---|---|---|---|
| | $MCC_{gt}$ | $MCC_{self}$ | $MCC_{gt}$ | $MCC_{self}$ |
| Synthetic Linear | 0.66 | 0.68 | **0.88** | **0.92** |
| Synthetic NonLinear | 0.55 | 0.56 | **0.71** | **0.85** |
| Synthetic Noise | 0.62 | 0.58 | **0.87** | **0.88** |

maximum linear correlations up to a permutation of components. We compute two types of MCC values. The first, denoted as $MCC_{gt}$, measures the MCC value between the learned and ground truth hidden confounders $Z$. We report the average MCC value of ten different runs. A high $MCC_{gt}$ value indicates successful recovery of the true hidden confounders. The second, denoted as $MCC_{self}$, involves training the same model multiple times and calculating the MCC for every pair of learned hidden confounders. Specifically, let $Z^{(1)}, \cdots, Z^{(N)}$ be the learned hidden confounders of different runs. We compute the MCC value for every pair $Z^{(i)}, Z^{(j)}$ and report the average MCC value. A high $MCC_{self}$ value indicates that the learned hidden confounders converge to a unique solution. We evaluate our model's identifiability using three synthetic datasets. Our model is compared against a modified version, denoted as 'GraphCFF w/o GMM.' Specifically, 'GraphCFF w/o GMM' doesn't use GMM as the prior distribution, so the identifiability of 'GraphCFF w/o GMM' cannot be guaranteed. The results in Table 2 demonstrate that our model achieves $MCC_{self}$ over $0.85$ for all three datasets, indicating stable learned hidden confounders. On the synthetic linear dataset, our model achieves an $MCC_{gt}$ value of $0.88$, showing its capability to accurately recover hidden confounders. Furthermore, GraphCFF outperforms 'GraphCFF w/o GMM' in terms of both $MCC_{gt}$ and $MCC_{self}$ values, indicating the importance of GMM prior.

## 6 Conclusions

In this work, we propose a method, known as GraphCFF, to address the fairness issue of GNNs by augmenting graph data with counterfactual generation. We introduce a causal model with hidden confounders on graphs, which considers the existence of hidden confounders affecting both node features and graph structures. We use an identifiable graph VAE model with a Gaussian mixture prior distribution to recover counterfactuals, enabling the learning of counterfactually fair downstream classifiers. Through experimental results, we demonstrate that our method can significantly improve counterfactual fairness while not sacrificing much accuracy. Furthermore, theoretical analysis and empirical results provide convincing evidence for the ability of our method to accurately identify hidden confounding factors.

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

# A PROOF OF IDENTIFIABILITY

**Theorem 2.** Let $F$ be the generation function of our proposed causal model with hidden confounders on graphs, defined as $F(\boldsymbol{Z}, S) = (\boldsymbol{A}, \boldsymbol{X}) = (F_A(\boldsymbol{Z}, S), F_X(\boldsymbol{Z}, F_A(\boldsymbol{Z}, S), S))$. If $F$ is weakly injective, then we have $P(\boldsymbol{Z}, C)$ to be identifiable from $P(\boldsymbol{A}, \boldsymbol{X}, S)$ up to permutation, scaling, and/or translation.

*Proof.* The proof of this Theorem is done in the following three steps.

In the first step, we aim to show that the problem of identifying the joint distribution $P(\boldsymbol{Z}, C)$ from $P(\boldsymbol{A}, \boldsymbol{X}, S)$ can be reduced to identifying $P(\boldsymbol{Z})$ from $P(\boldsymbol{A}, \boldsymbol{X}, S)$. This reduction is based on the well-known fact that finite Gaussian mixture distributions are identifiable up to permutations of the Gaussian components (Yakowitz & Spragins, 1968; Teicher, 1963). Specifically, since finite mixtures of Gaussian distributions are identifiable up to a permutation of the Gaussian components, we can recover both the mixture weights $P(C)$ and the corresponding Gaussian components $P(\boldsymbol{Z}|C)$. This implies that $P(\boldsymbol{Z}, C)$ can be identified from $P(\boldsymbol{Z})$. Therefore, it is sufficient to prove that $P(\boldsymbol{Z})$ is identifiable from $P(\boldsymbol{A}, \boldsymbol{X}, S)$ up to permutation, scaling, and/or translation.

In the second step, we show that $P(\boldsymbol{Z})$ is identifiable from $P(\boldsymbol{A}, \boldsymbol{X}, S)$ up to an affine transformation of $\boldsymbol{Z}$. It is proved by Theorem 7.

In the third step, we show that using the assumption that $\boldsymbol{Z}$ are independent latent factors, then $P(\boldsymbol{Z})$ is identifiable from $P(\boldsymbol{A}, \boldsymbol{X}, \boldsymbol{S})$ up to permutation, scaling, and/or translation of $\boldsymbol{Z}$. Since $P(\boldsymbol{Z})$ is identifiable up to an affine transformation by the second step, the claim follows from Theroem 8.

□

## A.1 IDENTIFIABILITY OF LATENT FACTORS

**Lemma 3.** Let $\boldsymbol{Z} = [Z_1, Z_2, \ldots, Z_n] \in \mathbb{R}^{n \times q}$ be the matrix of latent factors, where each $Z_i \in \mathbb{R}^q$. The latent factors of different nodes are independent of each other. Suppose that for each $Z_i$, the probability distribution follows the same Gaussian Mixture Model (GMM) with $K$ components and diagonal covariance matrices:

$$P(Z_i) = \sum_{k=1}^{K} \pi_k \mathcal{N}(Z_i | \mu_k, \boldsymbol{\Sigma}_k), \tag{8}$$

where $\pi_k$ is the mixture weight for the $k$-th Gaussian distribution, $\mu_k$ is the mean vector, and $\boldsymbol{\Sigma}_k = \mathrm{diag}(\sigma_{k,1}^2, \sigma_{k,2}^2, \ldots, \sigma_{k,q}^2)$ is the diagonal covariance matrix.

Then, when we consider the whole matrix $\boldsymbol{Z}$ as a vector of size $n \times q$ (reshaped from the matrix), the joint distribution $P(\mathrm{vec}(\boldsymbol{Z}))$ can be expressed as a Gaussian Mixture Distribution.

*Proof.* Consider the reshaped matrix $Z_{\mathrm{vec}} = [Z_1^T, Z_2^T, \cdots, Z_n^T]^T$. Let's compute the joint probability $P(Z_{\mathrm{vec}})$:

$$P(Z_{\mathrm{vec}}) = P(Z_1, Z_2, \ldots, Z_n) = \prod_{i=1}^{n} P(Z_i). \tag{9}$$

Since $\boldsymbol{\Sigma}_k$ is diagonal, we can express the Gaussian distribution as a product of $q$ univariate Gaussian distributions:

$$\mathcal{N}(Z_i | \boldsymbol{\mu}_k, \boldsymbol{\Sigma}_k) = \prod_{j=1}^{q} \mathcal{N}(Z_{ij} | \mu_{kj}, \sigma_{kj}^2). \tag{10}$$

Now, substituting this expression for $P(Z_i)$:

$$P(Z_{\text{vec}}) = \prod_{i=1}^{n} \left[ \sum_{k=1}^{K} \pi_k \prod_{j=1}^{q} \mathcal{N}(Z_{ij}|\mu_{kj}, \sigma_{kj}^2) \right]. \tag{11}$$

Now, let's look at one entry of $Z_{\text{vec}}$, say $Z_{uw}$, which is the element in the $u$-th row and $w$-th column of $\boldsymbol{Z}$. Its probability distribution can be written as:

$$P(Z_{uw}) = \sum_{k=1}^{K} \pi_k \mathcal{N}(Z_{uw}|\mu_{kw}, \sigma_{kw}^2). \tag{12}$$

Since each entry $Z_{uw}$ is independent, the joint distribution of $Z_{\text{vec}}$ is indeed a product of Gaussian Mixture Distributions, and each term is a mixture of univariate Gaussian distributions:

$$P(Z_{\text{vec}}) = \prod_{i=1}^{n}\prod_{j=1}^{q} P(Z_{ij}) = \prod_{i=1}^{n}\prod_{j=1}^{q} \left[ \sum_{k=1}^{K} \pi_k \mathcal{N}(Z_{ij}|\mu_{kj}, \sigma_{kj}^2) \right]. \tag{13}$$

Thus, when the covariance matrices are diagonal, the joint distribution of the reshaped matrix $Z_{\text{vec}}$ can be expressed as a single Gaussian mixture distribution. $\qquad\square$

**Lemma 4.** Let $F$ be the generation function of our proposed causal model with hidden confounders on graphs, defined as $F(\boldsymbol{Z}, S) = (\boldsymbol{A}, \boldsymbol{X}) = (F_A(\boldsymbol{Z}, S), F_X(\boldsymbol{Z}, F_A(\boldsymbol{Z}, S), S))$. We have $F$ is a piecewise affine function.

*Proof.* When extra inputs are used to combine piecewise affine functions, the resulting function remains a piecewise affine function. These additional inputs can be used to determine which linear segment or piece to use for a particular input value. In other words, if $F$ is a piecewise affine function for every fixed $S$, then $F$ is indeed a piecewise affine function. Therefore, it is sufficient to show that, when $S$ is fixed, $F$ is a piecewise affine function.

We first consider the output $\boldsymbol{A}$. Recall that $\boldsymbol{A}$ is computed as

$$\boldsymbol{A}_{ij} = \begin{cases} 1, & \text{if } s_i \neq s_j \text{ and } \text{sim}(Z_i, Z_j) > 1 - P_{\text{inter}} \\ 1, & \text{if } s_i = s_j \text{ and } \text{sim}(Z_i, Z_j) > 1 - P_{\text{intra}} \quad \text{for } i, j = 1, \cdots, n, \\ 0, & \text{otherwise} \end{cases} \tag{14}$$

where $\text{sim}(\cdot)$ computes the similarity between latent factors of two nodes. $P_{\text{inter}}$ and $P_{\text{intra}}$ are two constants that represent the intra-connection and inter-connection thresholds, respectively. For fixed $S$, $\boldsymbol{A}$ is determined by applying a comparison operation to the similarity measure $\text{sim}(Z_i, Z_j)$ against thresholds $1 - P_{\text{inter}}$ and $1 - P_{\text{intra}}$. Within each piece of the domain, defined by the conditions, the output $A_{ij}$ is constant. Therefore, $\boldsymbol{A}$ is a piecewise constant function with respect to $\boldsymbol{Z}$ when $S$ is held constant, which can be considered a special case of a piecewise affine function.

Next, we consider the output $\boldsymbol{X}$. Recall that $\boldsymbol{X}$ is computed as

$$X_i = F_{X_1}(Z_i, s_i) + \text{AGG}(\{F_{X_2}(Z_j, s_j) : j \in N_i\}) \quad \text{for } i = 1, \cdots, n, \tag{15}$$

where $N_i$ denotes the set of the nodes connected to the node $i$ in the graph, $\text{AGG}(\cdot)$ denotes an aggregation function, and $F_{X_1}$ and $F_{X_2}$ are two piecewise affine transformation functions. Then $F_{X_1}$ and $F_{X_2}$ divide the input space to multiple intervals $r_1, r_2, \cdots$. In the interval $r_1$, $F_{X_1}$ and $F_{X_2}$ are affine functions and can be denoted as $F_{X_1}(Z, s) = \boldsymbol{W}_1 Z + B_1 s + B_1'$ and $F_{X_2}(Z, s) = \boldsymbol{W}_2 Z + B_2 s + B_2'$ respectively, where $\boldsymbol{W}_1$ and $\boldsymbol{W}_2$ are some matrices, and $B_1$, $B_1'$, $B_2$, and $B_2'$ are some vectors. Then we have $X_i = \boldsymbol{W}_1 Z_i + B_1 s_i + B_1' + \text{AGG}(\{\boldsymbol{W}_2 Z_j + B_2 s_j + B_2' : j \in N_i\})$. If we consider each possible configuration of $\boldsymbol{A}$ as defining a different 'piece' of the domain, then within each piece, the aggregation function, such as mean operation, is a linear function. Therefore, each element of $\boldsymbol{X}$ can be considered as the result of a piecewise affine function, implying that the generation of matrix $\boldsymbol{X}$ as a whole can be considered a piecewise affine process.

Therefore, we conclude that the overall function $F$ can be considered a piecewise affine function.

$\qquad\square$

**Lemma 5.** Consider a pair of finite GMMs

$$P = \sum_{j=1}^{J} \lambda_j \mathcal{N}(\mu_j, \Sigma_j) \quad \text{and} \quad P' = \sum_{j=1}^{J'} \lambda'_j \mathcal{N}(\mu'_j, \Sigma'_j). \tag{16}$$

Assume that there exists a ball $B(x_0, \delta)$ such that $P$ and $P'$ induce the same measure on $B(x_0, \delta)$. Then $P \equiv P'$, i.e., $J = J'$ and for some permutation $\tau$ we have $\lambda_i = \lambda'_{\tau(i)}$ and $(\mu_i, \Sigma_i) = (\mu'_{\tau(i)}, \Sigma'_{\tau(i)})$.

*Proof.* We first show that the GMMs are real analytic functions. This step is straightforward. It is well known that the multivariate Gaussian density function is a real analytic function, and a finite sum of real analytic functions is also real analytic. Therefore, both $P$ and $P'$ are real analytic functions.

Next, we use the identity theorem for real analytic functions in our case. The identity theorem for real analytic functions states that if a real analytic function is equal to another real analytic function on an open subset of its domain, then the two functions are equal everywhere on their common domain. In our case, $P$ and $P'$ are equal on the ball $B(x_0, \delta)$. By the identity theorem, we can conclude that $P \equiv P'$.

In the last step, we need to show that the Gaussian components are the same between $P$ and $P'$. First, we argue that $J = J'$. Since $P$ and $P'$ are equal everywhere, the number of Gaussian components must be the same, because each component contributes uniquely to the overall shape of the GMM. Next, assume without loss of generality that $\lambda_1 \geq \lambda_2 \geq ... \geq \lambda_J$ and $\lambda'_1 \geq \lambda'_2 \geq ... \geq \lambda'_{J'}$. We argue that $\lambda_j = \lambda'_j$ for all $j$. Suppose for contradiction that there exists some $j$ such that $\lambda_j \neq \lambda'_j$. Then there must exist $k \neq j$ such that $\lambda_k \neq \lambda'_k$, because these mixture weights are themselves probabilities that satisfy the constraints $\sum_{i=1}^{J} \lambda_i = 1$ and $\lambda_i > 0$. Without loss of generality, we assume that $\lambda_j > \lambda'_j$. Then we have $\lambda'_k > \lambda_k$. However, in the neighborhood of the mode of the $k$-th Gaussian component, $P'$ should be greater than $P$, which contradicts the fact that $P = P'$ everywhere. Finally, we show that for each $j$, $(\mu_j, \Sigma_j) = (\mu'_j, \Sigma'_j)$. This is due to the fact that the mean and covariance completely characterize a Gaussian distribution, and two Gaussians that are equal everywhere must have the same mean and covariance.

This completes the proof. We have shown that if two GMMs induce the same measure on a ball, then they must be identical, up to a permutation of the components. $\qquad\square$

**Theorem 6.** (Kivva et al., 2022) Let $f, g$ be two piecewise affine functions that are weakly injective. Let $Z \sim \sum_{i=1}^{J} \lambda_i \mathcal{N}(\mu_i, \Sigma_i)$ and $Z' \sim \sum_{j=1}^{J'} \lambda'_j \mathcal{N}(\mu'_j, \Sigma'_j)$ be a pair of GMMs. Suppose that $f(Z)$ and $g(Z')$ are equally distributed. Then there exists an invertible affine transformation $h$ such that $h(Z) \equiv Z'$, i.e., $J = J'$ and for some permutation $\tau \in S_J$ we have $\lambda_i = \lambda'_{\tau(i)}$ and $h_\sharp \mathcal{N}(\mu_i, \Sigma_i) = \mathcal{N}(\mu'_{\tau(i)}, \Sigma'_{\tau(i)})$.

Here, we outline the key idea of the proof. Initially, given that both piecewise affine functions $f$ and $g$ demonstrate are weak injective, it can be inferred that $f$ and $g$ are invertible in a neighborhood of a specific point. Following this, based on Lemma 5, it can be demonstrated that there exists an invertible affine transformation between $Z$ and $Z'$. See complete proof of this theorem in Appendix A of Kivva et al. (2022). Theorem 6 implies that a mixture model whose components are piecewise affine transformations of a Gaussian is identifiable.

**Theorem 7.** Assume that $(\boldsymbol{Z}, \boldsymbol{A}, \boldsymbol{X}, S)$ are distributed according to our proposed causal model with hidden confounders on graphs. If the generation function $F$ is weakly injective, then $P(\boldsymbol{Z})$ is identifiable from $P(\boldsymbol{A}, \boldsymbol{X}, S)$ up to an affine transformation.

*Proof.* Assume we have two models, including (1) $(\boldsymbol{Z}, \boldsymbol{A}, \boldsymbol{X}, S)$ and a geneartion function $F$, and (2) $(\boldsymbol{Z'}, \boldsymbol{A'}, \boldsymbol{X'}, S')$ and a geneartion function $F'$. If two models are well trained, then we have $P(\boldsymbol{A'}, \boldsymbol{X'}|S') = P(\boldsymbol{A}, \boldsymbol{X}|S)$ and $P(S) = P(S')$. In other words, $F(\boldsymbol{Z})$ and $F'(\boldsymbol{Z'})$ have the same distribution. According to lemma 3, we know $\boldsymbol{Z}$ and $\boldsymbol{Z'}$ are a pair of GMMs. Moreover, by lemma 4, $F$ and $F'$ are piecewise affine functions. Therefore, by Theorem 6, there exists an

invertible affine transformation $h$ such that $h(\boldsymbol{Z}) \equiv \boldsymbol{Z}'$. In other words, $P(\boldsymbol{Z})$ is identifiable up to an invertible affine transformation. $\qquad\square$

## A.2 IDENTIFIABILITY UP TO A PERMUTATION, SCALING, AND TRANSLATION

**Theorem 8.** (Kivva et al., 2022) Let $J \geq 2$, and $\lambda_k > 0$ for $k = 1, \cdots, J$. Let $Z$ be given by

$$Z \sim \sum_{k=1}^{J} \lambda_k \mathcal{N}(\mu_k, \Sigma_k) \tag{17}$$

Assume that $\Sigma_k$ is diagonal for $k = 1, \cdots, J$. Let $Y = WZ + B$ for some matrix $W$ and some vector $B$. Moreover, assume that there exist indices $i_1, i_2 \in [J]$, such that all numbers $((\Sigma_{i_1})_{tt} / (\Sigma_{i_2})_{tt} \mid t \in [m])$ are distinct. Given $Y$, one can recover an invertible linear map $W'$, such that $(W')^{-1}W = QD$, where $Q$ is a permutation matrix and $D$ is a diagonal matrix with positive entries.

See the complete proof of this Theorem in Appendix E of Kivva et al. (2022).

Consider $h$ as an invertible affine transformation, and let $\boldsymbol{Z}'$ be defined as $\boldsymbol{Z}' \equiv h(\boldsymbol{Z})$. By Theorem 8, we have that $\boldsymbol{Z}$ can be recovered from $\boldsymbol{Z}'$ up to a permutation, scaling, and translation. In other words, given that every covariance matrix $\Sigma_k$ is diagonal, we can extend the conclusion that $\boldsymbol{Z}$ can be recovered up to an affine transformation to a more refined conclusion where $\boldsymbol{Z}$ can be recovered subject to a permutation, scaling, and translation.

## B ELBO

In this section, we provide detailed derivations of ELBO.

$$
\begin{aligned}
\log P(\boldsymbol{A}, \boldsymbol{X}|S) =& \mathbb{E}_{Q(\boldsymbol{Z}|\boldsymbol{A},\boldsymbol{X},S)}\left[\log \frac{P(\boldsymbol{A}, \boldsymbol{X}, \boldsymbol{Z}|S)}{P(\boldsymbol{Z}|\boldsymbol{A},\boldsymbol{X},S)}\right] \\
=& \mathbb{E}_{Q(\boldsymbol{Z}|\boldsymbol{A},\boldsymbol{X},S)}\left[\log \frac{P(\boldsymbol{A}, \boldsymbol{X}, \boldsymbol{Z}|S)Q(\boldsymbol{Z}|\boldsymbol{A},\boldsymbol{X},S)}{P(\boldsymbol{Z}|\boldsymbol{A},\boldsymbol{X},S)Q(\boldsymbol{Z}|\boldsymbol{A},\boldsymbol{X},S)}\right] \\
\geq& \mathbb{E}_{Q(\boldsymbol{Z}|\boldsymbol{A},\boldsymbol{X},S)}\left[\log \frac{P(\boldsymbol{A}, \boldsymbol{X}, \boldsymbol{Z}|S)}{Q(\boldsymbol{Z}|\boldsymbol{A},\boldsymbol{X},S)}\right] \\
=& \mathbb{E}_{Q(\boldsymbol{Z}|\boldsymbol{A},\boldsymbol{X},S)}\left[\log \frac{P(\boldsymbol{A}, \boldsymbol{X}|\boldsymbol{Z}, S)P(\boldsymbol{Z}|S)}{Q(\boldsymbol{Z}|\boldsymbol{A},\boldsymbol{X},S)}\right] \\
\overset{(a)}{=}& \mathbb{E}_{Q(\boldsymbol{Z}|\boldsymbol{A},\boldsymbol{X},S)}\left[\log \frac{P(\boldsymbol{A}|\boldsymbol{Z}, S)P(\boldsymbol{X}|\boldsymbol{A}, \boldsymbol{Z}, S)P(\boldsymbol{Z}|S)}{Q(\boldsymbol{Z}|\boldsymbol{A},\boldsymbol{X},S)}\right] \\
=& \mathbb{E}_{Q(\boldsymbol{Z}|\boldsymbol{A},\boldsymbol{X},S)}[\log P(\boldsymbol{A}|\boldsymbol{Z}, S) + \log P(\boldsymbol{X}|\boldsymbol{A}, \boldsymbol{Z}, S) \\
& + \log P(\boldsymbol{Z}|S) - \log Q(\boldsymbol{Z}|\boldsymbol{A},\boldsymbol{X},S)] \\
\overset{(b)}{=}& \mathbb{E}_{Q(\boldsymbol{Z}|\boldsymbol{A},\boldsymbol{X},S)}[\log P(\boldsymbol{A}|\boldsymbol{Z}, S) + \log P(\boldsymbol{X}|\boldsymbol{A}, \boldsymbol{Z}, S) \\
& + \log P(\boldsymbol{Z}) - \log Q(\boldsymbol{Z}|\boldsymbol{A},\boldsymbol{X},S)]
\end{aligned}
$$

Equation (a) holds because of the factorization of joint probability $P(\boldsymbol{Z}, \boldsymbol{A}, \boldsymbol{X}, S) = P(\boldsymbol{Z}, S)P(\boldsymbol{A}|\boldsymbol{Z}, S)P(\boldsymbol{X}|\boldsymbol{A}, \boldsymbol{Z}, S)$, and Equation (b) holds due to the independence between sensitive attribute $S$ and latent factors $\boldsymbol{Z}$.

$$
\begin{aligned}
\log P(\boldsymbol{A}, \boldsymbol{X}|S) =& \mathbb{E}_{Q(\boldsymbol{Z}, C|\boldsymbol{A}, \boldsymbol{X}, S)}\left[\log \frac{P(\boldsymbol{A}, \boldsymbol{X}, \boldsymbol{Z}, C|S)}{P(\boldsymbol{Z}, C|\boldsymbol{A}, \boldsymbol{X}, S)}\right] \\
=& \mathbb{E}_{Q(\boldsymbol{Z}, C|\boldsymbol{A}, \boldsymbol{X}, S)}\left[\log \frac{P(\boldsymbol{A}, \boldsymbol{X}, \boldsymbol{Z}, C|S)Q(\boldsymbol{Z}, C|\boldsymbol{A}, \boldsymbol{X}, S)}{P(\boldsymbol{Z}, C|\boldsymbol{A}, \boldsymbol{X}, S)Q(\boldsymbol{Z}, C|\boldsymbol{A}, \boldsymbol{X}, S)}\right] \\
\geq& \mathbb{E}_{Q(\boldsymbol{Z}, C|\boldsymbol{A}, \boldsymbol{X}, S)}\left[\log \frac{P(\boldsymbol{A}, \boldsymbol{X}, \boldsymbol{Z}, C|S)}{Q(\boldsymbol{Z}, C|\boldsymbol{A}, \boldsymbol{X}, S)}\right] \\
=& \mathbb{E}_{Q(\boldsymbol{Z}, C|\boldsymbol{A}, \boldsymbol{X}, S)}\left[\log \frac{P(\boldsymbol{A}, \boldsymbol{X}|\boldsymbol{Z}, S, C)P(\boldsymbol{Z}, C|S)}{Q(\boldsymbol{Z}, C|\boldsymbol{A}, \boldsymbol{X}, S)}\right] \\
=& \mathbb{E}_{Q(\boldsymbol{Z}, C|\boldsymbol{A}, \boldsymbol{X}, S)}\left[\log \frac{P(\boldsymbol{A}, \boldsymbol{X}|\boldsymbol{Z}, S)P(\boldsymbol{Z}, C|S)}{Q(\boldsymbol{Z}, C|\boldsymbol{A}, \boldsymbol{X}, S)}\right] \\
=& \mathbb{E}_{Q(\boldsymbol{Z}, C|\boldsymbol{A}, \boldsymbol{X}, S)}\left[\log \frac{P(\boldsymbol{A}|\boldsymbol{Z}, S)P(\boldsymbol{X}|\boldsymbol{A}, \boldsymbol{Z}, S)P(\boldsymbol{Z}, C|S)}{Q(\boldsymbol{Z}, C|\boldsymbol{A}, \boldsymbol{X}, S)}\right] \\
=& \mathbb{E}_{Q(\boldsymbol{Z}, C|\boldsymbol{A}, \boldsymbol{X}, S)}[\log P(\boldsymbol{A}|\boldsymbol{Z}, S) + \log P(\boldsymbol{X}|\boldsymbol{A}, \boldsymbol{Z}, S) + \log P(\boldsymbol{Z}, C|S) \\
& - \log Q(\boldsymbol{Z}|\boldsymbol{A}, \boldsymbol{X}, S) - \log Q(C|\boldsymbol{A}, \boldsymbol{X}, S)] \\
=& \mathbb{E}_{Q(\boldsymbol{Z}, C|\boldsymbol{A}, \boldsymbol{X}, S)}[\log P(\boldsymbol{A}|\boldsymbol{Z}, S) + \log P(\boldsymbol{X}|\boldsymbol{A}, \boldsymbol{Z}, S) + \log P(\boldsymbol{Z}, C) \\
& - \log Q(\boldsymbol{Z}|\boldsymbol{A}, \boldsymbol{X}, S) - \log Q(C|\boldsymbol{A}, \boldsymbol{X}, S)] \\
=& \mathbb{E}_{Q(\boldsymbol{Z}, C|\boldsymbol{A}, \boldsymbol{X}, S)}[\log P(\boldsymbol{A}|\boldsymbol{Z}, S) + \log P(\boldsymbol{X}|\boldsymbol{A}, \boldsymbol{Z}, S) + \log P(\boldsymbol{Z}|C) \\
& + \log P(C) - \log Q(\boldsymbol{Z}|\boldsymbol{A}, \boldsymbol{X}, S) - \log Q(C|\boldsymbol{A}, \boldsymbol{X}, S)]
\end{aligned}
$$

## C    MORE DISCUSSIONS

### C.1    FAIRNESS-AWARE GRAPH AUGMENTATIONS

Fairness-aware graph data augmentation is an essential strategy in mitigating bias in graph neural networks (GNNs). A primary issue in fairness is that the bias in training datasets is a significant source of discriminative behavior in models (Mehrabi et al., 2021; Olteanu et al., 2019). GNN models may inherit or even amplify bias from training data (Dai & Wang, 2021). This process can result in potentially discriminatory predictions, adversely impacting groups defined by sensitive attributes such as race and gender. Hence, Fairness-aware graph data augmentation, which reduces bias in the training data, emerges as a highly effective strategy for enabling models to learn fair representations. Spinelli et al. (2021) propose that the existence of edges connecting nodes with identical sensitive attributes can lead to prediction discrimination. To tackle this issue, they propose a biased edge drop algorithm that mitigates this tendency in graphs, thereby enhancing fairness in prediction tasks. Kose & Shen (2022) study correlations between sensitive attributes and learned node representations. They propose a series of graph augmentations aimed at minimizing the upper bounds of these correlations, resulting in improving fairness in GNNs. While previous methods rely on manually discovered properties of fair graph data in order to design fixed strategies, Ling et al. (2023b) propose learning fair representations via automated graph data augmentations. Their proposed Graphair can automatically identify and implement fairness-aware augmentations from input graphs, circumventing sensitive information while preserving other useful information. However, a significant shortcoming of these fairness-aware augmentation methods is that they ignore causal relationships in the augmented data. This omission could lead to a situation where the augmented data does not adhere to realistic scenarios. For instance, in order to achieve fair prediction outcomes for both females and males, these methods may generate a lot of tall and heavy female samples, a distribution that does not accurately reflect real-world demographics. This problem highlights the importance of incorporating causality into fairness-aware data augmentation strategies to ensure the creation of realistic and representative datasets for training GNNs.

### C.2    IDENTIFIABILITY OF LATENT-VARIABLE MODELS

Deep latent-variable models, such as VAEs, have been extensively used for unsupervised learning of latent representations. However, traditional results from nonlinear Independent Component Anal-

ysis (ICA) reveal a fundamental issue in VAEs, namely the lack of identifiability, where different latent variables can generate the same observed variables (Hyvärinen & Pajunen, 1999). This issue hinders the approximation of the true joint distribution over observed and latent variables. A recent breakthrough by Khemakhem et al. (2020a) demonstrates that the identifiability of VAES can be achieved up to certain nonlinear equivalences when side information is available. Building upon this idea, several methods have been proposed to achieve identifiability with some form of auxiliary information (Khemakhem et al., 2020b; Mita et al., 2021; Yang et al., 2021; Li et al., 2019; Sorrenson et al., 2020). For example, Yang et al. (2021) proposes to incorporate a Structural Causal Model (SCM) into a VAE model. During training, the true causal variables are used as labels, which guide the learning process to disentangle the latent representations and encourage the model to learn a meaningful and identifiable latent representation. Furthermore, Ahuja et al. (2022; 2021) propose methods that reduce the reliance on additional information. They show that if the underlying mechanisms guiding the evolution of latent variables are known, the latent variables can be identified up to the equivariances of these mechanisms. In contrast, some approaches avoid using additional forms of supervision and aim to enforce identifiability in a purely unsupervised setting (Kivva et al., 2022; Wang et al., 2021). These works demonstrate that identifiability is achievable under commonly adopted assumptions, indicating that auxiliary information may not be necessary. Another line of research focuses on restricting the selection of the variational family and prior distribution to improve identifiability (Kumar & Poole, 2020).

### C.3 Relation with Previous Studies about Counterfactual Fairness on Graphs

In comparison to prior research on counterfactual fairness in graphs, our causal model is more general. For example, NIFTY (Agarwal et al., 2021) considers a simple scenario where sensitive attributes are independent of both node features and the adjacency matrix. This situation is merely a special case within our broader setting. Specifically, if both functions $F_X$ and $F_A$ remain unchanged with respect to $S$, then sensitive attributes will not influence node features or the adjacency matrix. Similarly, GEAR (Ma et al., 2022) considers the scenario where $S$ is the only confounder for both $A$ and $X$. This is also a special case within our broader setting. Specifically, if both functions $F_X$ and $F_A$ are invariant with respect to $Z$, then our causal model is equivalent to GEAR's scenario.

Moreover, we consider one of the significant challenges in graph data, namely the non-independence of nodes. This implies that modifications to one node could potentially impact another. In contrast, prior studies assume node features cannot be influenced by their neighbors. This more simplistic view is yet another specific instance within our broader framework. Specifically, if $F_{X_2}$ in Eq. (3) consistently returns zero, then the node features remain unaffected by their neighbors. To further validate our approach, we provide experimental results, detailed in Appendix E.2, which demonstrate its effectiveness even in this specific scenario.

### C.4 Assumption about Independence Between Hidden Confounders and Sensitive Attributes

Our primary objective is to derive trustworthy counterfactuals. Importantly, we can obtain accurate counterfactuals without the explicit computation of latent factors influenced by $S$. This means that we can produce consistent counterfactuals using just $S$ and $Z$. Let $Z'$ denote those latent factors influenced by $S$. Two causal relationships need to be considered.

1. In the causal graph, $Z'$ is solely a child of sensitive attribute $S$, meaning $Z'$ is only causally affected by $S$. In this case, the value of $Z'$ can be expressed as $Z' = F_{Z'}(S)$, where $F_{Z'}(\cdot)$ describes how $Z'$ is dependent on $S$. The generation process of observed graph data can be expressed as $(A, X) = F(S, Z, Z')$, where $F$ corresponds to the generating mechanism from $S, Z, Z'$. Given $Z' = F_{Z'}(S)$, we have $(A, X) = F(S, Z, Z') = F(S, Z, F_{Z'}(S)) = F'(S, Z)$, where $F'$ is an alternative function. This implies that the generation process can be expressed only using $S$ and $Z$. Therefore, our proposed method can generate identical counterfactuals, as $F(S', Z, F_{Z'}(S')) = F'(S', Z)$.

2. In the causal graph, $Z'$ is the child of both sensitive attribute $S$ and other latent factors independent of $S$. This implies $Z'$ is causally affected by both $S$ and $Z$. The value of $Z'$ can be expressed as $Z' = F_{Z'}(S, Z)$, where $F_{Z'}(\cdot)$ describes how $Z'$ is dependent on both $S$ and $Z$. The generation process of observed graph data can be expressed as $(A, X) = F(S, Z, Z')$, where $F$

corresponds to the generating mechanism of graph data from $S, \boldsymbol{Z}, \boldsymbol{Z'}$. Given $\boldsymbol{Z'} = F_{Z'}(S, \boldsymbol{Z})$, we have $(\boldsymbol{A}, \boldsymbol{X}) = F(S, \boldsymbol{Z}, \boldsymbol{Z'}) = F(S, \boldsymbol{Z}, F_{Z'}(S, \boldsymbol{Z})) = F'(S, \boldsymbol{Z})$, where $F'$ is an alternative function. In this scenario as well, our proposed method can generate the same counterfactuals, as $F(S', \boldsymbol{Z}, F_{Z'}(S', \boldsymbol{Z})) = F'(S', \boldsymbol{Z})$.

In summary, even though there exist latent factors that can be affected by sensitive attributes, independence between $\boldsymbol{Z}$ and $S$ is reasonable. This assumption suffices to recover reliable counterfactuals.

### C.5 Assumption about Gaussian Mixture Prior of Hidden Confounders

While existing works (Madras et al., 2018; Ma et al., 2022; Louizos et al., 2017; Foster et al., 2022) use a normal distribution as the prior distribution of latent factors, we use GMM as the prior distribution of hidden confounders. It's essential to note that a GMM with a single Gaussian component is equivalent to a normal distribution. This makes the normal distribution a special case of GMM. Therefore, our framework offers a more generalized approach compared to existing works. Moreover, GMM is a universal approximator of densities (Goodfellow et al., 2016; Carreira-Perpinan, 2000; Scott, 2015), meaning it can represent a wide variety of continuous density functions. With enough mixture components, any smooth density can be approximated by a GMM to a specific nonzero level of accuracy. This highlights the adaptability of GMM in capturing various density distributions. In other words, our model can recover hidden confounders with diverse distributions, implying its superior applicability in real-world scenarios compared to existing works.

## D   More Details on Experimental Settings

### D.1   Datasets

We create three synthetic datasets, including Synthetic Linear, Synthetic NonLinear, and Synthetic Noise. For all three datasets, we generate a graph with $n = 1000$ nodes. We use a Gaussian mixture model with $K = 5$ components to generate the hidden confounders $\boldsymbol{Z}$. For each Gaussian component, each component of mean vector $\mu_k$ take a value between $-10$ and $10$, and covariance matrix $\Sigma_k$ is a diagonal matrix with diagonal values equal to $0.1$. The sensitive attribute for each node is sampled from a Bernoulli distribution as $S_i \sim \text{Bernoulli}(p_s)$, where $p_s = 0.5$ is the probability of $S_i = 1$. Afterwards, we generate the adjacency matrix $\boldsymbol{A}$ and node feature $\boldsymbol{X}$ using Eqs. (2) and (3), respectively. Node labels are generated in the same way as node features and then mapped to binary values. To be more specific, we generate the adjacency matrix as follows,

$$\boldsymbol{A}_{ij} = \begin{cases} 1, & \text{if } s_i \neq s_j \text{ and } \text{sim}(Z_i, Z_j) > 1 - P_{\text{inter}} \\ 1, & \text{if } s_i = s_j \text{ and } \text{sim}(Z_i, Z_j) > 1 - P_{\text{intra}} \\ 0, & \text{otherwise} \end{cases} \quad \text{for } i, j = 1, \cdots, n,$$

where we use cosine similarity as the $\text{sim}(\cdot)$ function and set $P_{\text{inter}} = 5 \times 10^{-4}, P_{\text{intra}} = 5 \times 10^{-3}$. We use mean pooling as the aggregation function in all three datasets. In Synthetic Linear, we set functions $F_{X_1}$ and $F_{X_2}$ as two linear transformations. To create a linear transformation, we generate a full-rank matrix. Specifically, we initialize a matrix, denoted $\boldsymbol{W}$, wherein each element is independently sampled from a standard normal distribution $\mathcal{N}(0, 1)$. Then we check the determinant of the matrix $\boldsymbol{W}$. If the determinant is zero, indicating that the matrix is singular, we repeat the previous step to generate a new matrix until a non-singular matrix is obtained. The resulting matrix $\boldsymbol{W}$ represents the desired linear transformation. In Synthetic NonLinear, we set $F_{X_1}$ and $F_{X_2}$ as two non-linear transformations. These transformations are achieved by using randomly initialized MLPs. Specifically, we use three-layer MLPs with Leaky-ReLU as the activation function. In Synthetic Noise, we introduce randomness to the adjacency matrix entries and add exponential noise to node features. Specifically, we use a Bernoulli distribution with a probability of $0.05$ to decide whether each entry in the adjacency matrix should be flipped. Furthermore, we add noise $\epsilon_i$ to each node's feature, where $\epsilon_i$ is sampled from the exponential distribution as $\epsilon_i \sim \exp(0.2)$.

For the real-world datasets, we use German, Credit, and Bail. The German dataset contains 1000 nodes, with each node representing a client of a German bank. The edges between these nodes represent similarities between their corresponding credit accounts. The Credit dataset consists of

Table 3: Comparisons between our method and baselines on node classification tasks in terms of ROC AUC, F1 score, DP, and EO. The best results are shown in bold.

| Dataset | Method | ROC AUC ↑ | F1 ↑ | DP ↓ | EO ↓ |
|---|---|---|---|---|---|
| Synthetic Linear | No augmentation | $95.96 \pm 0.39$ | $88.75 \pm 2.09$ | $10.55 \pm 0.94$ | $\mathbf{12.02 \pm 3.49}$ |
| | NIFTY | $\mathbf{96.26 \pm 1.62}$ | $\mathbf{89.14 \pm 0.79}$ | $11.18 \pm 1.89$ | $15.72 \pm 1.75$ |
| | GEAR | $94.20 \pm 1.82$ | $86.61 \pm 2.05$ | $5.90 \pm 2.85$ | $13.26 \pm 5.01$ |
| | GraphCFF | $95.02 \pm 0.76$ | $85.40 \pm 0.44$ | $\mathbf{2.05 \pm 1.70}$ | $13.29 \pm 3.11$ |
| Synthetic NonLinear | No augmentation | $\mathbf{93.27 \pm 1.18}$ | $90.76 \pm 1.02$ | $20.21 \pm 2.05$ | $9.72 \pm 1.96$ |
| | NIFTY | $92.51 \pm 0.90$ | $91.86 \pm 0.56$ | $14.20 \pm 1.80$ | $5.56 \pm 1.36$ |
| | GEAR | $86.30 \pm 3.86$ | $87.62 \pm 1.57$ | $7.76 \pm 3.04$ | $\mathbf{3.89 \pm 2.75}$ |
| | GraphCFF | $88.17 \pm 1.60$ | $\mathbf{92.58 \pm 0.57}$ | $\mathbf{5.77 \pm 1.98}$ | $3.89 \pm 1.42$ |
| Synthetic Noise | No augmentation | $\mathbf{97.35 \pm 0.43}$ | $\mathbf{93.45 \pm 1.41}$ | $4.43 \pm 2.18$ | $5.56 \pm 1.57$ |
| | NIFTY | $96.47 \pm 0.55$ | $93.18 \pm 0.74$ | $\mathbf{1.82 \pm 1.14}$ | $3.33 \pm 1.70$ |
| | GEAR | $91.35 \pm 6.02$ | $87.91 \pm 4.15$ | $2.83 \pm 1.22$ | $2.22 \pm 1.57$ |
| | GraphCFF | $95.40 \pm 0.96$ | $91.55 \pm 0.46$ | $3.21 \pm 0.95$ | $\mathbf{1.11 \pm 1.36}$ |
| German | No augmentation | $\mathbf{64.68 \pm 2.26}$ | $78.52 \pm 1.41$ | $16.96 \pm 6.58$ | $14.18 \pm 6.88$ |
| | NIFTY | $63.79 \pm 3.40$ | $78.54 \pm 1.79$ | $6.19 \pm 2.06$ | $7.25 \pm 5.67$ |
| | GEAR | $63.81 \pm 1.38$ | $81.67 \pm 1.57$ | $6.56 \pm 2.77$ | $5.83 \pm 2.03$ |
| | GraphCFF | $61.56 \pm 2.22$ | $\mathbf{82.35 \pm 1.81}$ | $\mathbf{2.15 \pm 2.04}$ | $\mathbf{1.45 \pm 2.05}$ |
| Credit | No augmentation | $\mathbf{75.93 \pm 1.27}$ | $87.97 \pm 1.13$ | $1.91 \pm 1.60$ | $1.03 \pm 2.20$ |
| | NIFTY | $75.91 \pm 0.67$ | $\mathbf{88.13 \pm 0.78}$ | $1.36 \pm 0.49$ | $1.27 \pm 0.49$ |
| | GEAR | $73.42 \pm 2.13$ | $88.03 \pm 2.43$ | $0.75 \pm 0.89$ | $0.74 \pm 0.44$ |
| | GraphCFF | $72.36 \pm 1.12$ | $87.20 \pm 0.73$ | $\mathbf{0.24 \pm 0.34}$ | $\mathbf{0.41 \pm 0.58}$ |
| Bail | No augmentation | $\mathbf{96.45 \pm 1.18}$ | $\mathbf{91.54 \pm 1.32}$ | $6.76 + 1.51$ | $2.26 \pm 1.02$ |
| | NIFTY | $96.19 \pm 0.50$ | $91.53 \pm 0.70$ | $6.15 \pm 0.34$ | $\mathbf{1.49 \pm 1.33}$ |
| | GEAR | $89.05 \pm 1.09$ | $83.35 \pm 1.01$ | $6.69 \pm 0.74$ | $2.54 \pm 1.02$ |
| | GraphCFF | $88.97 \pm 1.38$ | $82.83 \pm 1.85$ | $\mathbf{5.53 \pm 1.23}$ | $2.07 \pm 0.77$ |

$30,000$ nodes, with each node representing an individual with features such as education, credit history, and age. The edges between these nodes represent the similarities in spending and payment patterns among these individuals. Bail dataset includes $18,876$ nodes, with each node representing a defendant who was released on bail from U.S. state courts between 1990 and 2009. The edges between these nodes represent the similarities in their criminal histories and demographics.

## D.2 Implementation Details

For a fair comparison, we use the same classification model to evaluate all methods. For the classification model, we use a GCN model. The number of GCN layers is two, and we use a global mean pooling as the readout function. We set the hidden size as 16. The activation function is ReLU. We use the Adam optimizer (Kingma & Ba, 2015) to train the classification model with $1 \times 10^{-4}$ learning rate and $1 \times 10^{-4}$ weight decay.

## E More Experimental Results

### E.1 More Evaluation Metrics

In this section, we provide experimental results with more evaluation metrics. Beyond merely assessing accuracy, we incorporate the F1 score and ROC AUC as additional performance evaluation metrics, offering a more comprehensive evaluation of model performance. To provide a comprehensive evaluation of fairness, we also include two metrics that are commonly used in statistical fairness, namely demographic parity (DP) and equal opportunity (EO). DP quantifies the disparity in positive prediction rates between two distinct groups and is mathematically expressed as $\Delta_{DP} = |\mathbb{P}(\hat{Y} = 1|S = 0) - \mathbb{P}(\hat{Y} = 1|S = 1)|$. On the other hand, EO focuses on fairness concerning true positive rates and is defined as $\Delta_{EO} = |\mathbb{P}(\hat{Y} = 1|S = 0, Y = 1) - \mathbb{P}(\hat{Y} = 1|S = 1, Y = 1)|$. It's worth noting that group fairness is different from counterfactual fairness. While group fairness metrics, such as DP and EO, provide valuable insights into potential group-level biases, they don't directly measure counterfactual fairness.

Table 4: Comparisons between our method and baselines on Synthetic 0-hop dataset. The best results are shown in bold.

| Synthetic 0-hop | ACC $\uparrow$ | ROC AUC $\uparrow$ | CF $\downarrow$ |
|---|---|---|---|
| No augmentation | **73.67 $\pm$ 2.05** | 77.84 $\pm$ 3.11 | 16.20 $\pm$ 2.56 |
| NIFTY | 70.67 $\pm$ 1.25 | 75.65 $\pm$ 0.63 | 13.13 $\pm$ 1.54 |
| GEAR | 72.67 $\pm$ 2.05 | 76.46 $\pm$ 0.63 | 7.60 $\pm$ 2.66 |
| GraphCFF | 72.00 $\pm$ 1.41 | **79.21 $\pm$ 1.21** | **5.33 $\pm$ 1.35** |

We conduct experiments on all six datasets using performance metrics ROC AUC and F1 score, as well as fairness metrics DP and EO. Results in Table 3 show that our proposed GraphCFF has comparable performance compared to 'No augmentation' in terms of both ROC AUC and F1 score in the majority of scenarios. Furthermore, even though DP and EO don't directly measure counterfactual fairness, our GraphCFF generally demonstrates a reduction in prediction bias in terms of DP and EO. Overall, our GraphCFF can significantly improve fairness while not sacrificing much performance of the model. It is worth noting that in some cases, our approach not only enhances fairness but also improves performance. For example, in the Synthetic NonLinear dataset, our GraphCFF outperforms the 'No augmentation' with a higher F1 score and lower DP and EO.

### E.2 MORE EXPERIMENTS ON SYNTHETIC DATASETS

To verify our discussion in Appendix C.3, we conduct additional experiments to demonstrate the effectiveness of our method in the specific scenario that node features cannot be influenced by their neighbors. As we mentioned in Appendix C.3, the independence between node features can be achieved by setting $F_{X_2}$ to zero functions. Specifically, we generate a new synthetic dataset, namely Synthetic 0-hop. The generation process of Synthetic 0-hop is the same as Synthetic Linear, except that $F_{X_2}$ is a zero function rather than a full-rank linear transformation in this case. The results in Table 4 show that our GraphCFF outperforms all baselines in terms of counterfactual fairness and ROC AUC. This means that scenarios, where node features remain uninfluenced by their neighbors, are specific cases within our broader setting. Consequently, our proposed GraphCFF proves to be adept at handling these special cases.

