# OpenReview forum: "Counterfactual Fairness on Graphs: Augmentations, Hidden Confounders, and Identifiability"
_ICLR.cc/2024/Conference — Submitted to ICLR 2024_

### Official Review · Reviewer_GkRy · 2023-10-29

**Soundness:** 3 good
**Presentation:** 3 good
**Contribution:** 2 fair
**Rating:** 5
**Confidence:** 3

**Summary:**

This paper presents a novel method for attaining counterfactual fairness for graph neural networks by enhancing the graph data with counterfactual generation. The proposed method considers hidden confounders on graphs and uses an identifiable graph VAE model to simultaneously estimate these confounders and learn the generation functions of the causal model. By enforcing consistency in the predictions of classifiers for different counterfactual graphs, the authors achieve graph counterfactual fairness in these classifiers. The effectiveness of the proposed method is demonstrated through experiments on various graph tasks. The author also provides a theoretical justification in the Appendix.

**Strengths:**

1, The paper addresses an important and timely problem of achieving fairness on graphs, which is becoming increasingly important in many real-world applications.

2, The proposed method is based on a solid theoretical foundation of causal inference and graph VAE models, which provides a principled and interpretable approach to achieving counterfactual fairness on graphs.

3, The experimental results demonstrate the effectiveness of the proposed method in improving counterfactual fairness on various graph tasks, which suggests its potential for practical applications.

**Weaknesses:**

1, In section 3.1, the author introduces the property of the graph data that the change of the node features of one node may alter the other. However, in the following sections 3.2, 3.3, 3.4 and the experimental sections, the author does not specify how or how effectively their introduced method can capture this kind of knock-on property. In addition, on generating the node features, the author assumes the a node’s feature only depends on its one-hop neighbors, which is far from the scenarios in the reality.

2, In section 3.2, the author holds the assumptions that the confounder Z and the sensitive attribute are identically independently distributed. It is better to provided experimental results to justify this property. Eg providing the correlation plot or some statistical testing  results showing that these two variables following different distributions .

**Questions:**

This study introduces a counterfactual fairness evaluation metric, as shown in Equation 7. This metric restricts flipping a single node's sensitive attribute at a time. In contrast, in paper (Ma, Jing, et al. 2022), the metric is formulated to accommodate the flipping of multiple nodes, leading to a wider range of counterfactual scenarios. My concern is that the proposed metric may lack diversity in its counterfactual samples, potentially limiting its ability to provide a comprehensive assessment of counterfactual fairness.

Reference:
Ma, Jing, et al. "Learning fair node representations with graph counterfactual fairness." Proceedings of the Fifteenth ACM International Conference on Web Search and Data Mining. 2022.

---

### Official Review · Reviewer_t9GV · 2023-10-31

**Soundness:** 2 fair
**Presentation:** 2 fair
**Contribution:** 2 fair
**Rating:** 5
**Confidence:** 4

**Summary:**

The paper introduces GraphCFF, a model designed to achieve counterfactual fairness in Graph Neural Networks (GNNs) with the incorporation of counterfactual data augmentation. This paper considers the presence of unobserved confounders that have a causal impact on both node features and graph features. Additionally, the authors present an identifiability result for the Variational Autoencoder (VAE) model with a Gaussian mixture prior distribution. Experimental results on synthetic datasets and real-world data demonstrate the method's effectiveness to some extent.

**Strengths:**

- Achieving counterfactual fairness in GNNs with counterfactual data augmentation is an existing relevant problem and it is important to consider the existence of the unobserved confounders that causally affect both node features and graph features.

- The paper is well-written, making it easy to understand and follow the proposed method and its results.

**Weaknesses:**

- This paper focuses on generating counterfactual graphs, but it lacks an evaluation of the quality of these generated counterfactuals. It would be beneficial to see a comparison between the counterfactuals produced by the proposed method and ground-truth counterfactuals on synthetic datasets, or a comparison of how well downstream classifiers perform based on the generated counterfactuals and the ground-truth counterfactuals.

- The identifiability result assumes of weakly injective. It would be valuable to provide a discussion on the strength of this assumption, highlighting its implications and potential limitations.

- The paper mentions that the method selects a node $i$ and generates a counterfactual graph during each training step. However, it does not specify how many training steps are needed to obtain a sufficient sample size for a good downstream classifier, which is an important practical consideration.

- The paper lacks clarity on how to train downstream classifiers using the generated counterfactuals. Do the counterfactuals need to be split into training/validation/test sets, and if so, how should this be done?

- For real-world datasets, the paper does not clearly explain how the authors generated the counterfactual data based on causal discovery methods. I checked the Appendix D.1 but cannot find the answer.

**Questions:**

- There is an inaccuracy in the statement in section 3.1: "we explicitly consider the causal relationship between nodes, meaning that the node features of node $i$ can be influenced by its neighbours." The text seems to describe a correlation, instead of causal relationship. This should be corrected for precision.

---

### Official Review · Reviewer_7FQE · 2023-10-31

**Soundness:** 3 good
**Presentation:** 4 excellent
**Contribution:** 3 good
**Rating:** 5
**Confidence:** 3

**Summary:**

The paper considers counterfactual fairness on graphs. According to them (I haven't checked), they are the first to consider a full causal model on graphs: protected attributes and hidden latent variables are independent but cause the edges in the graph and node features. The more general framework allows for considering graphs that are generated as a result of latent variables and protected attributes e.g. social networks where people are friends based on their demographic background and personality.

Both the generation function and distribution of latent variables are unknown. They learn both simultaneously using the ELBO method. I hadn't seen this before and enjoyed looking it up, thanks!

They use a mixture of Gaussians to model the hidden cofounders. This seems like a fine assumption to me but I'm skeptical that it leads to the "identifiability" property they want; basically, they want only one set of parameters to give the right distribution of the observables so, if they fit the observables well, they know they have the right latent distribution. They show a theoretical result that a weakly injective function gives this property for their construction (I didn't look at the proof).

They also compare their method to several others on synthetic and real data. Generally, their method achieves better counterfactual fairness but sometimes sacrifices accuracy. It's hard to compare the results because they're generally incomparable (how do you compare higher accuracy but lower counterfactual fairness?).

Edit 11/22/23: Increasing review to 4 to be more in line with other reviewers average of 5. I don't see any responses from authors so no change to my substantive evaluation.

Edit: I don't see 4 as an option so it's a 5 from me.

**Strengths:**

• The paper considers a novel graph counterfactual fairness problem. I'm generally inclined to think of all the counterfactual fairness extensions as not particularly useful. However, the setting you consider where hidden variables cause the graph structure seems very applicable and practical.

• The paper uses the ELBO method which seems perfectly applicable for the problem. It seems like the contribution here is that observable variables are now the adjacency matrix and node features.

• The paper considers the problem of identifiability which seems particularly relevant for recovering the right latent variables. They show a theoretical result which proves if the function is weakly injective then it satisfies their identifiability condition.

**Weaknesses:**

• I'm generally skeptical about the identifiability analysis. Your main theorem seems intuitively obvious: if a function is (approximately) injective then of course only one set of parameters will give each output.

• I'm confused why generating the latent variable as a mixture of Gaussians would help with identifiability. In your experiments, I assume you generate the true latent variables with a mixture of Gaussians? Then of course a model that also generates latent variables according to a mixture of Gaussians will get a better fit than a model that doesn't. If I understand what's going on here correctly, your presentation seems borderline misleading. I also want a theoretical reason for why a mixture of Gaussians helps with identifiability. Your theoretical result only says that the function being (approximately) injective suffices; you could probably have chosen any other distribution for the latent variables and this would still be true.

• I find it concerning that you don't apply the contrastive learning component of the NIFTY and GEAR approaches. If you don't apply the entire method, then it seems unfair to say yours performs better.

• My understanding is that the real world data you use does not have a graph structure. I would love to see experiments on real data with a graph structure like social networks.

**Questions:**

Does the real data have a graph structure? If not, what do you do to make your model applicable?

For the identifiability evaluation, how do you know the ground truth latent variables? Are they generated from a synthetic process where the latent variables are a mixture of Gaussians?

Why do you limit the number of nodes for dependence in the graph generating process? Independence after a distance of 1 seems small and arbitrary.

If my concerns are addressed in the discussion, I would be willing to increase my rating to 5 or 6.

---

### Official Review · Reviewer_QRDh · 2023-11-02

**Soundness:** 3 good
**Presentation:** 3 good
**Contribution:** 3 good
**Rating:** 5
**Confidence:** 3

**Summary:**

This work studies the research problem of counterfactual fairness on graphs. The authors argued several weaknesses of existing solutions and proposed GraphCFF with better designs such as more reasonable causal DAG with unobservable latent confounders Z.

**Strengths:**

1. The paper is overall clearly written and easy to follow.
2. This work proposed a much more realistic causal model than the ones in baselines, namely having Z -> A, Z-> X, and A -> X in fig 1c.
3. In addition to the basic VAE, the authors also included GMM to better model the latent factors.
4. The proposed method showed better fairness-accuracy trade-off than baselines.

**Weaknesses:**

1. Though the proposed causal model is more realistic than the ones in baselines, obvious constrains such as S has to be binary still exist. As the real-life examples the authors mentioned in line 5 of Sec. 2 are not binary.
2. With the multiple component design, the proposed model looks quite heavy and doesn't seem to be scalable. I'd appreciate some complexity analysis as well as runtime comparisons.
3. In addition to the above point, the datasets used in the experiments are very small. I'd suggest the authors to also conduct experiments on larger datasets to better demonstrate the usefulness of the proposed method.

**Questions:**

please refer to the weaknesses

---

### Meta-Review · Area_Chair_rgqX · 2023-12-05

**Metareview:**

The paper provides some novel ideas for counterfactual fairness on graphs. However, reviewers pointed out some unclear passages regarding the results, and no discussion was started. Given this gap, I believe it is better for now that the paper takes into account the numerous suggestions from the reviewers for a new iteration.

**Justification For Why Not Higher Score:**

Unclear answers to questions posed.

**Justification For Why Not Lower Score:**

N/A

---

### Decision · Program_Chairs · 2024-01-16

Reject